# Matrix Metalloproteinase 9 Induced in Esophageal Squamous Cell Carcinoma Cells via Close Contact with Tumor-Associated Macrophages Contributes to Cancer Progression and Poor Prognosis

**DOI:** 10.3390/cancers15112987

**Published:** 2023-05-30

**Authors:** Shuichi Tsukamoto, Yu-ichiro Koma, Yu Kitamura, Kohei Tanigawa, Yuki Azumi, Shoji Miyako, Satoshi Urakami, Masayoshi Hosono, Takayuki Kodama, Mari Nishio, Manabu Shigeoka, Hiroshi Yokozaki

**Affiliations:** 1Division of Pathology, Department of Pathology, Kobe University Graduate School of Medicine, Kobe 650-0017, Japan; stsuka@med.kobe-u.ac.jp (S.T.); y.kitamura-0916@people.kobe-u.ac.jp (Y.K.); 188m863m@gsuite.kobe-u.ac.jp (K.T.); 202m856m@gsuite.kobe-u.ac.jp (Y.A.); shoji224@med.kobe-u.ac.jp (S.M.); urasato@med.kobe-u.ac.jp (S.U.); takodama@med.kobe-u.ac.jp (T.K.); marin@med.kobe-u.ac.jp (M.N.); mshige@med.kobe-u.ac.jp (M.S.); hyoko@med.kobe-u.ac.jp (H.Y.); 2Division of Gastro-Intestinal Surgery, Department of Surgery, Kobe University Graduate School of Medicine, Kobe 650-0017, Japan; hosono.masayoshi@aijinkai-group.com; 3Division of Gastroenterology, Department of Internal Medicine, Kobe University Graduate School of Medicine, Kobe 650-0017, Japan

**Keywords:** esophageal squamous cell carcinoma, tumor-associated macrophage, direct co-culture, direct contact, matrix metalloproteinase 9, prognostic factor, interleukin-8, signal transducer, activator of transcription 3, immunohistochemistry

## Abstract

**Simple Summary:**

The crucial role of tumor-associated macrophages (TAMs) in the disease progression of various types of tumors, including esophageal squamous cell carcinoma (ESCC), has been well established. To investigate the close interplay between cancer cells and TAMs, we previously established a direct co-culture system. Our aim was to identify useful molecular markers whose expression is regulated by interactions with cancer cells and TAMs, to predict patient prognoses, or to gain a deeper understanding of the ESCC mechanism. Our method revealed that direct contact with macrophages stimulated matrix metalloproteinase 9 (MMP9) production in ESCC cells, and that MMP9 played an integral role in cancer cell migration and invasion in vitro. Moreover, MMP9 expression in ESCC cells was significantly correlated with high infiltration of TAMs and poor clinical outcomes in immunohistochemical analysis. Our study suggests the potential for TAMs or MMP9 as important prognostic markers or therapeutic ESCC targets.

**Abstract:**

Tumor-associated macrophages (TAMs) contribute to disease progression in various cancers, including esophageal squamous cell carcinoma (ESCC). We have previously used an indirect co-culture system between ESCC cell lines and macrophages to analyze their interactions. Recently, we established a direct co-culture system to closely simulate actual ESCC cell-TAM contact. We found that matrix metalloproteinase 9 (MMP9) was induced in ESCC cells by direct co-culture with TAMs, not by indirect co-culture. MMP9 was associated with ESCC cell migration and invasion, and its expression was controlled by the Stat3 signaling pathway in vitro. Immunohistochemical analyses revealed that MMP9 expression in cancer cells at the invasive front (“cancer cell MMP9”) was related to high infiltration of CD204 positive M2-like TAMs (*p* < 0.001) and was associated with worse overall and disease-free survival of patients (*p* = 0.036 and *p* = 0.038, respectively). Furthermore, cancer cell MMP9 was an independent prognostic factor for disease-free survival. Notably, MMP9 expression in cancer stroma was not associated with any clinicopathological factors or patient prognoses. Our results suggest that close interaction with TAMs infiltrating in cancer stroma or cancer nests induces MMP9 expression in ESCC cells, equipping them with more malignant features.

## 1. Introduction

Esophageal cancer (EC) is a highly aggressive gastrointestinal malignancy and is the sixth leading cause of cancer-related deaths globally. The incidence of EC varies significantly among populations and regions, with the predominant histological type being esophageal squamous cell carcinoma (ESCC) in Asia and Africa [1]. In Japan, the crude incidence rate was 20.9 per 100,000 people and the age-standardized incidence rate was 7.2 per 100,000 people in 2019, according to the latest EC statistics [2]. Despite advances in surgery, chemotherapy, and radiotherapy, the long-term prognosis of EC patients remains poor, with a reported 5-year survival rate of 59.9% among those who underwent esophagectomy in Japan [3]. The primary cause of this outcome is mainly attributed to the high incidence of local or lymph node metastatic recurrence [3,4].

A cancerous tissue consists of cancer cells and abundant stromal cells such as tumor-associated macrophages (TAMs) and cancer-associated fibroblasts (CAFs). The interaction between cancer cells and TAMs is known to facilitate tissue remodeling and cancer cell invasion, thereby promoting local progression or metastasis to lymph nodes or distant organs [5]. We previously reported that the infiltration of CD204-positive M2-like macrophages in ESCC tissues significantly correlated with poor clinical outcomes [6]. Through the use of an in vitro indirect co-culture system between ESCC cell lines and macrophages derived from peripheral blood monocytes (PBMos), we identified several factors that mediate their interaction, including growth differentiation factor (GDF) 15 [7], interleukin (IL)-8 [8], C-C motif chemokine ligand (CCL) 1 [9], and CCL3 [10].

In cancerous tissues, some cancer cells and TAMs have direct contact, which is difficult to replicate in traditional in vitro co-culture systems. To address this limitation, we recently established an in vitro direct co-culture system. To better understand the gene expression changes in this system, we performed a cDNA microarray analysis to compare gene expression between ESCC cells monocultured and co-cultured directly with macrophages (registered with accession number GSE174796) [11]. By comparing these data with our previously deposited data (GSE118642) [12], using the indirect co-culture system, we identified several genes that were upregulated only in the direct co-culture system. Among these genes, we previously reported S100 calcium-binding proteins A8 and A9 (S100A8/A9) [11] and IL-7 receptor (IL-7R) [13] as cancer-promoting factors. In the present study, we further analyzed the newly established direct co-culture system and focused on the role of matrix metalloproteinase (MMP) 9 in promoting ESCC.

## 2. Materials and Methods

### 2.1. Cell Lines

Three ESCC cell lines, TE-9 (derived from poorly differentiated ESCC), TE-10 (derived from well-differentiated ESCC), and TE-11 (derived from moderately differentiated ESCC), were obtained from the RIKEN BioResource Center (Tsukuba, Japan). These cell lines were maintained in RPMI-1640 medium (FUJIFILM Wako Pure Chemical, Osaka, Japan) containing 10% fetal bovine serum (FBS) (Sigma-Aldrich, St. Louis, MO, USA) and 1% antibiotic-antimycotic (Invitrogen, Carlsbad, CA, USA) [14]. The individuality of the three ESCC cell lines was confirmed by short tandem repeat analysis at RIKEN and the Cell Resource Center for Biomedical Research, Institute of Development, Aging and Cancer, Tohoku University (Sendai, Japan). These cell lines tested negative for mycoplasma using a Venor^®^ Gem Classic Mycoplasma Detection kit (Minerva Biolabs, Berlin, Germany).

### 2.2. Preparation of Peripheral Blood-Derived Macrophages

Peripheral blood mononuclear cells (PBMCs) were obtained from healthy volunteers using a previously established protocol [7]. CD14-expressing peripheral blood monocytes (PBMos) were labeled with anti-CD14 microbeads (130-050-201; Miltenyi Biotec, Bergisch Gladbach, Germany) and selectively collected from PBMCs by positive selection using an autoMACS^®^ Pro Separator (Miltenyi Biotec). The collected PBMos were then suspended in RPMI-1640 with 10% FBS and 25 ng/mL recombinant human (rh) M-CSF (R&D Systems, Minneapolis, MN, USA) and seeded in a 10-cm dish at 1 × 10^6^ seeding density for the direct co-culture, or in a 0.4-µm pore size transwell (BD Falcon, Lincoln Park, NY, USA) placed in a 6-well plate at 1 × 10^5^ seeding density for the indirect co-culture. The cells were cultured for six days to differentiate into macrophages.

### 2.3. Direct and Indirect Co-Culture Systems

Routinely maintained TE-9, TE-10, or TE-11 cells were trypsinized and washed with RPMI-1640 without FBS three times. For the direct co-culture system, TE-9 cells at 1 × 10^5^ cells/mL concentration, and TE-10 or TE-11 cells at 2 × 10^5^ cells/mL in RPMI-1640 without FBS were prepared; 10 mL suspensions (containing 1 × 10^6^ TE-9, 2 × 10^6^ TE-10, or 2 × 10^6^ TE-11 cells, respectively) were added to PBMo-derived macrophages that had been washed with RPMI-1640 without FBS three times. After two days of direct co-culture, the cells were detached by trypsinization and EpCAM-positive ESCC cells were positively selected using an autoMACS^®^ Pro Separator with anti-EpCAM microbeads (130-061-101; Miltenyi Biotec).

For the indirect co-culture system, 0.5 × 10^5^ cells/mL (TE-9) or 1 × 10^5^ cells/mL (TE-10 or TE-11) suspension in RPMI-1640 without FBS were prepared; 2 mL of suspensions (1 × 10^5^ TE-9, 2 × 10^5^ TE-10, or 2 × 10^5^ TE-11 cells) were cultured in a 6-well plate and the cells were allowed to adhere to the bottom surface. Following this, a 0.4-µm pore size transwell (BD Falcon) containing PBMo-derived macrophages that had been washed with RPMI-1640 without FBS three times and then supplemented with 1 mL of fresh RPMI-1640 without FBS was placed onto the ESCC cells. After two days of indirect co-culture, the transwells were discarded and the remaining ESCC cells were detached and used for analysis.

For both co-culture systems, ESCC cells cultured without co-culture with macrophages were set as the monocultured controls.

### 2.4. cDNA Microarray

We reviewed and re-analyzed our previously deposited cDNA microarray data (GSE174796) [11]. Gene ontology analysis and pathway analysis were performed by GeneSpring GX (Agilent Technologies, Inc., Santa Clara, CA, USA) [15].

### 2.5. Cytokine Array

After the direct co-culture, TE-11 cells resuspended in RPMI-1640 without FBS were seeded in a 6-cm dish at a density of 1.7 × 10^6^ cells/3 mL and cultured for 24 h. The culture supernatants were collected and centrifuged at 1000 rpm for 5 min and smaller contaminants were sedimented. The supernatants were filtered using a 0.2-µm filter and stored at −80 °C. Cytokine array analysis was performed according to the manufacturer’s instruction (#ARY005B, R&D systems). The arrays were captured using the ImageQuant^TM^ LAS 4000 mini (FUJIFILM, Tokyo, Japan), and the spot densities were measured using the ImageJ 1.53t software ver. 1.8.0 (National Institutes of Health, Bethesda, MD, USA).

### 2.6. Quantitative Real-Time PCR (qPCR)

Total RNA was extracted from cultured cells using the RNeasy Mini Kit (Qiagen, Hilden, Germany). The concentration of mRNA was measured using the NanoDrop Lite (Thermo Fisher Scientific, Waltham, MA, USA). To test expression changes of *GAPDH*, *MMP9*, and *STAT3* genes after the indirect or direct co-culture and *STAT3* mRNA knockdown, quantitative real-time PCR (qPCR) was performed. The primers used for *GAPDH* were: 5′-ACC ACA GTC CAT GCC ATC AC-3′ (forward) and 5′-TCC ACC ACC CTG TTG GCT GTA-3′ (reverse). The primers for *MMP9* were: 5′-ATG CGT GGA GAG TCG AAA TC-3′ (forward) and 5′-TAC ACG CGA GTG AAG GTG AG-3′ (reverse). The primers for *STAT3* were: 5′-GAG AAG GAC ATC AGG GGT AAG-3′ (forward) and 5′-AGT GGA GAC ACC AGG ATA TTG-3′ (reverse). SYBR green master mix (Applied Biosystems, Foster City, CA, USA) was used for qPCR amplification. For qPCR experiments after treatment with recombinant human proteins, the probes used were *MMP9*: Hs00234579_m1 (Applied Biosystems), and *GAPDH*: Hs07286624_g1 (Applied Biosystems). The ABI StepOnePlus Real-time PCR system (Applied Biosystems) with TaqMan Gene Expression Master Mix (Applied BioSystems) was also used. The expression of the target genes in the respective samples was normalized to the levels of *GAPDH*.

### 2.7. Western Blot

The cellular proteins from cultured cells were extracted by lysing in lysis buffer (50 mM Tris-HCl pH 7.5, 125 mM NaCl, 5 mM EDTA and 0.1% Triton X-100) containing 1% protease inhibitor and 1% protein phosphatase inhibitor cocktail (Sigma-Aldrich). Western blot was performed following previously reported methods [11]. Briefly, proteins were electrophoresed in a 5–20% sodium dodecyl sulfate-polyacrylamide gradient gel. After electrophoresis, proteins were transferred to the PVDF membrane with an iBlot^®^ Gel Transfer Stack (Invitrogen). The membrane was blocked using 5% skim milk and then incubated overnight at 4 °C with a primary antibody. After washing, the membrane was incubated with a secondary antibody for 90 min at 25 °C. Chemiluminescence was excited by ImmunoStar^®^ Reagents (FUJIFILM Wako Pure Chemical) and captured by the ImageQuant^TM^ LAS 4000 mini (FUJIFILM).

The primary antibodies used were phosphorylated (p)-Stat3 Tyr705 (#9145, Cell Signaling Technology; CST, Beverly, MA, USA), total (t)-Stat3 (#4904, CST), pAkt Ser473 (#4060, CST), tAkt (#9272, CST), pp38 MAPK (#4511, CST), tp38 MAPK (#8690, CST), CXCR1 (#MAB330, R&D Systems), CXCR2 (#ab65968, Abcam, Cambridge, UK), and β-actin (#4970, CST). The signal intensities of bands were quantified using the ImageJ 1.53t.

### 2.8. STAT3 mRNA Knockdown Using Small Interfering (si) RNA

In this experiment, siRNA was used to knock down the expression of *STAT3* mRNA in ESCC cells. Specifically, si-*STAT3* (sc-29493, Santa Cruz Biotechnology, Dallas, TX, USA) was used as the specific siRNA targeting *STAT3* mRNA, while MISSION^®^ siRNA Universal Negative Control #1 (NC, Sigma-Aldrich) was used as a negative control. Lipofectamine RNAiMAX (Invitrogen) was used to transfect siRNAs into the ESCC cells. The cells were then cultured in RPMI-1640 medium supplemented with 10% FBS for 2 days before the culture media containing siRNAs were removed. The cells were then passed to be halved in their numbers and cultured in RPMI-1640 without FBS for 2 days before being used for PCR and ELISA experiments.

### 2.9. Transwell Migration and Invasion Assays

1 × 10^5^ ESCC cells were resuspended in 300 µL of RPMI-1640 without FBS after the direct co-culture and transferred to transwell inserts with an 8-µm pore filter (BD Falcon) for migration assays and the inserts of a Corning^®^ BioCoat^®^ Invasion Chamber (Corning, Tewlbury, MA, USA) for invasion assays (Upper chambers); 800 µL of RPMI-1640 with 0.5% FBS (for assays of TE-9 or TE-10) or 0.1% FBS (for assays of TE-11) were placed in 24-well plates (Lower chambers). For assays of TE-9 or TE-10, 0.5 µM of MMP9 inhibitor (ab142180, Abcam) dissolved in dimethyl sulfoxide (DMSO) was added to the lower chambers. For assays of TE-11, 0.1 µM of MMP9 inhibitor was added to the lower chambers. Upper chambers were placed onto lower chambers and the cells were cultured for 48 h. Subsequently, the cells were fixed and stained using Diff-Quik^®^ (Sysmex, Kobe, Japan). Cells on the upper surface of the inserts were removed thoroughly using cotton swabs and cells that migrated or invaded the lower surface were observed under a microscope. We randomly captured five and four images in migration and invasion assays per insert at 200× magnification, respectively, and counted the number of cells. Relative migration or invasion was calculated by dividing the number of cells by that in the control (monocultured and without MMP9 inhibitor).

### 2.10. Enzyme Linked Immunosorbent Assay (ELISA)

ESCC cells were resuspended in RPMI-1640 without FBS and seeded at a density of 1.0 × 10^6^ cells/3 mL for TE-9 and TE-10 cells, and 1.7 × 10^6^ cells/3 mL for TE-11 cells in a 6-cm dish. The cells were cultured for 24 h and the culture supernatants were harvested as described in the cytokine array section. The Quantikine^®^ ELISA Human MMP9 (#DMP900, R&D Systems) or IL-8/CXCL8 (#D8000C, R&D Systems) immunoassay was used to measure the harvested culture supernatants. The culture supernatants of ESCC cells after *STAT3* knockdown or treatment with recombinant human proteins were also processed as described above and used for MMP9 ELISA. The optical densities were measured using the Microplate Reader Infinite^®^ 200 PRO (Tecan, Männedorf, Switzerland).

### 2.11. Gelatin Gel Zymography

Culture supernatants of ESCC cells after the co-culture were loaded onto polyacrylamide gel containing 1% of gelatin, using a gelatin gel zymography kit (AK47, Cosmo Bio, Tokyo, Japan). After electrophoresis, the gels were then soaked in enzyme reaction buffer for 24 or 48 h at 37 °C to enable gelatinolytic reaction. After protein staining and discoloration, the bands formed by gelatinolytic activity were identified. Gel images were captured and band density was analyzed using the ImageJ 1.53t.

### 2.12. ESCC Cell Treatment by Recombinant Human Interleukin-8 (rhIL-8) or S100-Calcium Binding Protein A8/A9 (rhS100A8/A9)

Recombinant human interleukin-8 (rhIL-8) and S100-calcium binding protein A8/A9 (S100A8/A9) were purchased from R&D Systems and BioLegend (San Diego, CA, USA) respectively. TE-9, TE-10, or TE-11 cells were resuspended in RPMI-1640 without FBS at a density of 5 × 10^5^/4 mL and seeded in 6-cm dishes. After incubation for 24 h, the culture medium was replaced by 4 mL of FBS-free RPMI-1640 alone (control) or supplemented with recombinant protein (100 ng/mL of rhIL-8 or 10 µg/mL of rhS100A8/A9). The ESCC cells were further incubated for 24 h and then used for experiments.

### 2.13. Phosphoinositide 3-Kinase (PI3K) or p38 Mitogen-Activated Protein Kinase (MAPK) Inhibition Assays

TE-9, TE-10, or TE-11 cells suspended in RPMI-1640 supplemented with 10% FBS were seeded at a density of 5 × 10^5^/5 mL dish in a 6-cm dish. Once the cells attached to the bottom, the culture medium was replaced with 3 mL of RPMI-1640 without FBS alone (control) or with LY294002 (PI3K inhibitor, 10 µM; CST) for Akt pathway inhibition or with SB203580 (p38 MAPK inhibitor, 10 µM; CST) for p38 inhibition. After 24 h, the culture medium was collected and used for MMP9 ELISA and the cells were lysed for mRNA extraction.

### 2.14. Tissue Samples and Immunohistochemistry

We used 69 human ESCC tissue samples that were surgically resected at the Kobe University Hospital (Kobe, Japan) from 2005 to 2010. None of the patients had undergone neoadjuvant chemotherapy or radiotherapy before their surgeries. Informed consent was obtained from all patients for the use of their resected samples for medical research. This study was conducted in accordance with the Declaration of Helsinki and approved by the Kobe University Institutional Review Board (B210103). All resected samples were fixed with 10% formalin, embedded in paraffin wax, sliced at a thickness of 4-µm, and histologically diagnosed. We reviewed and re-evaluated the diagnoses according to the 11th edition of the Japanese Classification of Esophageal Cancer [16,17] and the 8th edition of the Union for International Cancer Control (UICC) TNM Classification of Malignant Tumours [18].

Immunohistochemistry (IHC) was performed with EnVision™+ Dual Link System-HRP with 3,3′-diaminobenzidine (Dako Cytomation, Glostrup, Denmark). The primary antibody against MMP9 (1:100, CST) was utilized, and MMP9 expression on IHC was separately evaluated in cancer cells at the invasive front and in the stromal cells. In cancer cells, the definition of positive staining was stronger staining than surrounding fibroblasts or lymphocytes. Cancer cell expression was divided into negative and positive groups by scanning the invasive front. Cancer stromal expression was stratified into low- and high-groups based on the intensity compared to adjacent non-cancerous tissue. For IL-8 expressions, previous examinations were reused [8]. The evaluation and scoring were independently performed by three pathologists (authors S.T., Y.-i.K. and H.Y.).

### 2.15. Statistical Analysis

All experiments were performed in triplicates and repeated independently at least three times. The results are presented as the mean ± standard error (SEM), and the statistical significance was tested using a two-sided Student’s *t*-test. The correlation between clinicopathological parameters and immunohistochemical analysis results was assessed using a Chi-square test. Kaplan–Meier curves were used to illustrate overall and disease-free survival and statistical significance was determined using the log-rank test. Univariate and multivariate analyses were performed using the Cox proportional hazard regression model to determine the significance of the parameters. A *p*-value of <0.05 was defined as statistically significant. Statistical analyses were performed using the SPSS Statics ver. 22 (IBM, Chicago, IL, USA).

## 3. Results

### 3.1. Direct Co-Culture with Macrophages Induces the Secretion of Multiple Humoral Factors from TE-11 ESCC Cells, including MMP9 and IL-8

To explore the change in cytokine secretion profile caused by direct co-culture with macrophages, we tested the culture supernatants of mono- and co-cultured TE-11 cells using cytokine array analysis. After the direct co-culture, TE-11 cells secreted various humoral factors, more than their monocultured counterparts, including MMP9 and IL-8 (Figure 1A,B, Appendix A). Along with the cytokine array, we reviewed the cDNA microarray analysis previously reported by us (GSE174796) [11]. The microarray also revealed that MMP9 was upregulated and genes belonging to “Matrix Metalloproteinases” or “Degradation_of_the_extracellular_matrix” pathways by the direct co-culture in TE-11 cells (Appendix A). Thus, we chose to investigate the function and regulation of MMP9 in ESCC cells in greater detail, specifically focusing on their interaction with macrophages.

### 3.2. Direct Co-Culture with Macrophages Upregulates MMP9 Gene Transcription and Protein Secretion in ESCC Cells

The gene transcription of *MMP9* was examined in ESCC cell lines (TE-9, TE-10, and TE-11) by qPCR (Figure 2A). We found that *MMP9* was upregulated in all ESCC cells examined after the direct co-culture but downregulated in all ESCC cells after the indirect co-culture. In addition to the gene transcription changes, MMP9 protein secretion, measured with ELISA (Figure 2B), was significantly increased by the direct co-culture in all ESCC cells. In TE-9 and TE-11 cells, MMP9 secretion was decreased by the indirect co-culture; however, in TE-10 cells, it was increased. Gelatin zymography (Figure 2C) showed similar results to ELISA but also revealed that the gelatinolytic activity of secreted MMP9 was preserved in the culture supernatants. However, we were unable to detect the activated form of MMP9.

### 3.3. Direct Co-Culture with Macrophages Activates the Stat3 Signal Pathway in ESCC Cells, Resulting in MMP9 Upregulation

To investigate changes in altered signaling pathways, we performed a Western blot of Stat3, Akt and p38 MAPK. Among them, Stat3 phosphorylation was enhanced by the direct co-culture, while it was decreased or unchanged by indirect co-culture (Figure 3A and Appendix A). The regulation of Stat3 phosphorylation was consistent with that of MMP9 expression, leading us to focus on Stat3 in the present study. When *STAT3* expression was silenced using siRNA, *MMP9* mRNA was also significantly downregulated (Figure 3B). ELISA for MMP9 protein also demonstrated its downregulation at the protein level (Figure 3C). Additionally, inhibition of phosphoinositide 3-kinase (PI3K) or p38 suppressed MMP9 secretion in all ESCC cell lines used (Appendix A).

### 3.4. ESCC Cells Exhibit an Augmented Migration and Invasion following Direct Co-Culture with Macrophages through MMP9 Function

Next, we studied the effects of MMP9 on the phenotypes of ESCC cells. As MMP9 is known to participate in cell migration and invasion, we performed transwell migration and invasion assays (Figure 4 and Appendix A). Consistent with our previous findings [11], direct co-culture with macrophages resulted in enhanced ESCC cell migration (Figure 4A,B and Appendix A) and invasion (Figure 4C,D and Appendix A). Treatment with an MMP9 inhibitor (Figure 4 and Appendix A) significantly abrogated the increase in migration of TE-9 and TE-10 cells, and the increase in invasion of TE-9 and TE-11 cells. However, the migration of TE-11 cells was not affected by the MMP9 inhibitor. While the direct co-culture did not significantly increase the invasion of TE-10 cells, a trend towards increased invasion was observed. Furthermore, the MMP9 inhibitor appeared to diminish the increase in TE-10, albeit lacking statistical significance.

### 3.5. MMP9 Expression by Cancer Nests in ESCC Tissues Significantly Correlates with Aggressive Clinicopathological Factors and Poor Prognoses

To investigate MMP9 expression in actual human ESCC tissues, we conducted IHC to evaluate the intensity of staining. The expression of MMP9 in cancer nests was detected at the invasive front and divided into negative (50 out of 69 cases) and positive (19 out of 69 cases) (Figure 5A). Stromal cell expression was stratified into low (21 out of 69 cases) and high (48 out of 69 cases) based on the MMP9 expression of adjacent non-neoplastic tissue (Figure 5C). Kaplan–Meier analyses revealed that MMP9 expression in cancer nests was significantly correlated with overall (*p* = 0.036) and disease-free survivals (*p* = 0.038) (Figure 5B), whereas in stromal cells it was not significantly correlated with either (Figure 5D). Next, we studied the correlations between MMP9 expressions and clinicopathological factors or numbers of infiltrating macrophages. The evaluation of infiltrating macrophages was previously described [6]. Additionally, MMP9 expression in cancer nests was significantly associated with the depth of invasion (*p* = 0.038), and infiltration of CD68 (*p* = 0.012), CD163 (*p* = 0.004), and CD204 (*p* < 0.001) positive macrophages, whereas in stromal cells it was not correlated with any of the selected clinicopathological parameters and numbers of infiltrating macrophages (Table 1). Univariate analyses revealed that the depth of invasion (*p* < 0.001), lymphatic vessel invasion (*p* = 0.001), blood vessel invasion (*p* = 0.030), pathological stage (*p* < 0.001), infiltration of CD68 (*p* = 0.009), CD163 (*p* = 0.049), and CD204 (*p* = 0.002) positive macrophages, and cancer nest MMP9 positivity (*p* = 0.038) were significantly correlated with patient poor disease-free survivals (Table 2). Multivariate Cox regression analyses showed that not only the depth of invasion (*p* = 0.010) but also cancer nest MMP9 expression (*p* = 0.026) was an independent indicator of patient poor disease-free survival (Table 2).

### 3.6. Direct Co-Culture with Macrophages Also Promotes IL-8 Secretion from ESCC Cells and IL-8 Partially Induces MMP9 Production in ESCC Cells

In the cytokine array displayed in Figure 1A, we observed an increase in IL-8 secretion from TE-11 cells due to the direct co-culture. Furthermore, we confirmed that ESCC cells produced more IL-8 after the direct co-culture compared to their monocultured counterparts (Figure 6A). We hypothesized that the direct co-culture-induced IL-8 secretion had an effect on MMP9 upregulation in ESCC cells. After examining the expressions of known IL-8 receptors, CXCR1 and CXCR2 on ESCC cells (Figure 6B), we treated ESCC cell lines with rhIL-8. The rhIL-8 stimulated *MMP9* transcription in all ESCC cells used (Figure 6C) but induced MMP9 secretion only from TE-10 (Figure 6D). We observed upregulated Stat3 phosphorylation only in TE-9 after rhIL-8 treatment (Appendix A). Additionally, we examined the expressions of IL-8, CXCR1, and CXCR2, which we previously reported [8], and MMP9 expression of cancer nests in the human ESCC tissues. CXCR2 expression was significantly associated with cancer nest MMP9 expression (*p* = 0.010) whereas CXCR1 was not (*p* = 0.591) (Appendix A). The correlation between IL-8 positivity and cancer nest MMP9 positivity was close to statistical significance (*p* = 0.056, Appendix A).

### 3.7. S100A8/A9 Also Partially Induces MMP9 Production in ESCC Cells

As we previously reported, the direct co-culture induces S100A8/A9 production in ESCC cells and S100A8/A9 promotes ESCC cell migration and invasion. In the present study, we showed MMP9 upregulation and subsequent promotion of ESCC cell migration and invasion upon the direct co-culture but these phenotypes could be explained by an effect of S100A8/A9. To test whether MMP9 production is induced by S100A8/A9, we treated ESCC cells with rhS100A8/A9 and examined MMP9 transcription and secretion. Similar to the treatment with rhIL-8, *MMP9* gene transcription was increased in all ESCC cells used (Appendix A) but MMP9 secretion was promoted only from TE-10 cells (Appendix A).

## 4. Discussion

Matrix metalloproteinases (MMPs) are a family of zinc-dependent proteases that include several subgroups such as collagenases, gelatinases, stromelysins, matrilysins, and membrane types. These enzymes degrade the extracellular matrix (ECM) or process various bioactive substances to play crucial roles in tissue remodeling, angiogenesis, or progression of various diseases. While numerous roles of MMPs in cancers have been clarified, the complex regulation of their expressions and activities implies that many of their functions remain elusive [19,20,21]. In the case of ESCC, MMP1, 3, 7, 9, and 10 have all been reported to be involved in disease progression [22,23,24,25,26]. MMP9, also known as gelatinase B, is one of the most well-studied MMPs. The full-length MMP9 protein (proMMP9, about 92 kDa) consists of four major domains: the prodomain, catalytic domain, linker domain, and hemopexin domain [19,20,21]. After being secreted into extracellular spaces, the prodomain is removed by multiple proteases such as MMP3, 7, and 10, furin, or plasmin, and then MMP9 gains its proteolytic activity [19,20,21]. It catalyzes ECM components such as type IV collagen, laminin, and elastin, and therefore, is thought to be involved in cancer invasion or metastasis [20]. Although it has been reported that ESCC patients with high MMP9 expression have a poor prognosis [20], the novelty of this study lies in the fact that the interaction with macrophages induced MMP9 expression in ESCC cells.

In contrast to our previous reports [11,13], we conducted both indirect and direct co-culture experiments between ESCC cells and PBMo-derived macrophages. Our results showed that the indirect co-culture only induced MMP9 secretion in TE-10. Previous studies reported that indirect co-culture between cancer cells and macrophages could upregulate MMP9 production in various cancer cell lines, including ovarian carcinoma [27,28], head and neck carcinoma [29], ESCC [30], gastric carcinoma, and fibrosarcoma cell lines [31]. These findings suggest that humoral factor-mediated interactions between cancer cells and macrophages can promote MMP9 production in certain tumor cells. However, in one of these studies, MMP9 levels did not change in several cell lines after indirect co-culture [31]. We believe that the variability in our results from the indirect co-culture experiments is due to the heterogeneity of the cell lines used. In contrast, the direct co-culture results revealed that MMP9 production was increased in all ESCC cells tested. We hypothesize that the closer interaction of the cancer cells and macrophages in direct co-culture may utilize additional mechanisms such as surface molecule interactions [32] or extracellular vesicles [32,33]. Direct co-culture between cancer cells and mesenchymal stem cells has been reported to have led to MMP9 upregulation in prostate and head and neck cancer cells, but not in indirect co-culture [34,35].

Our previous reports have shown that multiple signaling pathways are activated in ESCC cells when directly co-cultured with macrophages [11,13]. In the present study, we focused on the Stat3 pathway because its phosphorylation was increased by the direct co-culture but conversely decreased by the indirect co-culture. This pattern was similar to the behavior of MMP9 regulation. The importance of the Stat3 pathway in the interaction between cancer cells and macrophages has been clarified in some previous studies. In glioma or ovarian cancer models, Stat3 was phosphorylated both in cancer cells and macrophages by direct co-culture and promoted cancer cell proliferation and production of IL-6 or IL-10 from the macrophages [32,36]. Another study described indirect co-culture using SKOV3 (ovarian carcinoma) cells and THP-1 derived macrophages highly enhanced IL-8 levels in the co-culture supernatant, which lead to Stat3 phosphorylation and then polarized macrophages into an M2-like phenotype and induced stem cell-like features in cancer cells [37]. We did not observe Stat3 phosphorylation by the indirect co-culture, which was likely due to differences in the cell lines used or the cellular interactions from the direct co-culture. We demonstrated that *STAT3* silencing considerably suppressed MMP9 transcription and secretion in ESCC cell lines. One study has investigated the intracellular Stat3-MMP9 axis, with HN4, HN6, and Cal27 (head and neck carcinoma) cells and showed that treatment with rhIL-8-induced Stat3 phosphorylation and MMP9 expression, which were counteracted by an addition of Stat3 phosphorylation inhibitor (cryptotanshinone) [29]. As IL-8 only induced phosphorylation of Stat3 in a single ESCC cell line, it is plausible that other factors may also be contributing to the observed phosphorylation of Stat3 in all ESCC cell lines that were co-cultured with macrophages. For instance, in SKOV3 cells, phosphorylated Stat3 was found to directly bind to the *MMP9* gene promoter and activate its transcription after stimulation with rhIL-6 [38]. In the present study, we did not identify any upstream factors that induce Stat3 activation in ESCC cells that directly interacted with macrophages. Akt and p38 pathways appeared to be involved in MMP9 secretion in basal-state ESCC cells. However, despite the increased phosphorylation of Akt in TE-11 and p38 in TE-10 and TE-11 cells, MMP9 secretion from these cells did not elevate. In the present co-culture systems, the Stat3 pathway explained MMP9 regulation best among the signaling pathways examined.

As previously reported, the direct co-culture promoted migration and invasion of TE-9, TE-10, and TE-11 cells. In the present study, we demonstrated that MMP9 partly contributed to this effect by using the MMP9 inhibitor ab142180, which is cell-permeable and can bind intra and extracellular MMP9. In a study on the migration of ESCC cell lines, similar to our results, the MMP9 inhibitor R-94138 suppressed the migration of TE-9 and -10 cells with MMP9 expressions but did not affect the migration of TE-5 cells without MMP9 expression [39]. As the transwell migration assay utilized a membrane without ECM-coating, it is possible that the effects of MMP9 on the migration of ESCC cell lines are independent of its catalytic activity. Additionally, there is a report on the non-catalytic function of MMP9 that affects cell migration. In this study, COS-1 cells (monkey kidney epithelial cells) were transfected with *MMP9* cDNA and exhibited increased migration compared to control cells. Notably, an MMP9 inhibitor CT1746 or the introduction of enzymatically inactive mutation to MMP9 (MMP9E^230^ → A) did not suppress the increased migration due to MMP9 expression. Additionally, depletion of the hemopexin domain showed a significant reduction in increased cell migration induced by MMP9 transfection [40], suggesting that the hemopexin domain’s affinity to cell surface molecules, such as CD44, LRP-1, Ku, or integrin [20,40,41,42], may be involved in cell migration signaling. The intracellular mechanisms involved in MMP9 transcription to secretion and their contribution to cell motility are still not fully understood. Our findings showed that the MMP9 inhibitor ab124180 did not affect TE-11 migration, indicating that TE-11 migration may not depend on MMP9-surface receptor interactions. However, in invasion assays using ECM-coated membranes, ab124180 significantly suppressed increased invasion of TE-9 and -11. Although the numbers of TE-10 invaded cells varied, overall trends were similar to TE-9 and -11, suggesting that these changes resulted from the catalytic activity of MMP9. In the present study, gelatin zymography results did not show the presence of active form MMP9 (~84 kDa) in the culture supernatants, which is consistent with previous reports on tissue samples or cell culture supernatants. For instance, human placenta [43] or ESCC tissues [24] and culture supernatants of MKN1 (gastric carcinoma) [31] and TE-7, -8, -9, and -10 (ESCC) cells [39] did not contain active form MMP9, although they all express proMMP9. However, it is known that proMMP9 bound to gelatin or type IV collagen can exhibit partial catalytic activity [43]. In the ECM-coated membrane used in the invasion assay, this process might have contributed to the observed effects.

Researchers have examined the effect of MMP9 expression in cancerous tissues on clinical outcomes using IHC. There are reports in the fields of ESCC [22,23,25], oral squamous cell [44], gastric [45], breast [46], and bladder [47] carcinomas that suggest MMP9 expression in cancer cells is significantly related to poor clinical outcomes including patient survival or lymph node metastases. Conversely, MMP9 expression in cancer cells and stroma both indicated better patient survival in uterine cervical carcinoma [48]. It appears controversial whether MMP9 contributes to better or worse clinical outcomes in ovarian carcinoma [49]. These reports suggest that MMP9 function in cancer tissues varies, serving either as a cancer activator or suppressor, depending on the type of cancer. Within our total 69 ESCC cases, we obtained 19 cases of MMP9 positive in cancer cells at the invasive front (referred to “cancer cell MMP9” below). Compared with other reports on ESCC, MMP9 expression in our study revealed a lower positivity rate [23,25], which we attribute to differences in the IHC protocol and evaluation method of IHC. Nonetheless, our findings are consistent with previous reports, which indicate that MMP9 expression in cancer cells is associated with aggressive cancer characteristics and participates in cancer cell motility and invasion in vivo. Notably, we found that MMP9 expression in stromal cells did not correlate with clinicopathologic parameters or patient survivals, despite previous evidence suggesting that MMP9 derived from cancer stroma contributes to cancer progression [5]. Stroma contains abundant inflammatory cells expressing MMP9 such as macrophages, specifically TAMs, or neutrophils, but we speculate that not all of them participate in the tumor microenvironment. Our examination of the correlation between cancer cell MMP9 and macrophage infiltration used previously reported data [6], and revealed a significant association, particularly with CD204-positive M2-like macrophages. This is consistent with our results of the direct co-culture experiment. We hypothesize that cancer cells and M2-like macrophages closely communicate at the invasive front, resulting in MMP9 expression. To our best knowledge, this is the first study to separately evaluate MMP9 expression in cancer nests and stroma and analyze its correlations with clinicopathological factors and patient survival. In a previous study, the infiltration of cancer stromal CD163-positive macrophages was examined in ESCC tissues. Higher infiltration of CD163 positive cells significantly correlated with MMP9 expression mainly in the stroma, microvessel densities, and worse patient prognoses [50]. Although these results may appear to contradict our findings, this study did not set strict boundaries between cancer nests and stroma. Cancer cells included in the observed area might contribute to poor clinical outcomes. Our study demonstrates that MMP9 expression in cancer cells leads to shorter overall and disease-free survival. Among the articles listed above, studies on ESCC [22,25], breast carcinoma [46], cervical carcinoma [48], and ovarian carcinoma [49] carried out survival analyses based on MMP9 expression profiles. Combined with these previous reports, we have further confidence that MMP9 in cancer cells is closely associated with ESCC progression. Our evaluation method does not use multiple stratified scores (such as low, moderate, or high in staining intensity). This simple method enables highly objective and reproducible analysis. We also revealed that cancer cell MMP9 is an independent prognostic factor of disease-free survival through multivariate Cox regression analysis. This is the first study on ESCC to report that cancer cell MMP9 is an independent prognostic factor in patient survival. Our study supported the importance of MMP9 in ESCC progression. MMP9 has been a promising therapeutic target and several MMP9 inhibitors were tested in clinical trials. Although some inhibitors showed suppressing effects in earlier-stage cancers, all of them finally failed to reach the market because of their adverse effects or administration difficulties [21]. Detailed mechanisms of production or activation of MMP9 (i.e., interaction with TAMs) will help create a better MMP9 inhibitor in the future.

On the cytokine array, we observed that TE-11 co-cultured with macrophages secreted more IL-8 compared to monocultured TE-11. Our previous report indicated that IL-8 was expressed not only in TAMs but also in cancer cells in ESCC. We showed that IL-8 expression in cancer nests significantly correlated with CD204 positive macrophage infiltration [8], suggesting that TAMs induced IL-8 expression and may have some relation to MMP9 expression in cancer cells. Almost all the cancer cell MMP9 positive cases expressed IL-8 (18/19) on IHC, and the correlation was close to being statistically significant (*p* = 0.056). In vitro, treatment with rhIL-8 for ESCC cells induced MMP9 secretion only in TE-10, though the effect was smaller than that observed in direct co-culture. A study on head and neck carcinoma reported that rhIL-8 enhanced MMP9 expression in cancer cells [28]; however, we consider that the response to rhIL-8 in vitro differs among cell types. CXCR1 and CXCR2 are both IL-8 receptors, and it is reported that CXCR2, not CXCR1, plays a role in inducing MMP9 in neutrophils [51]. We have demonstrated a significant correlation between CXCR2 expression and cancer cell MMP9 in ESCC tissues (*p* = 0.010) through IHC, and the expression ratio of CXCR2 to CXCR1 was the highest in TE-10 among the three ESCC cell lines on Western blot. Although we could not investigate this mechanism in the present study, a similar mechanism in neutrophils may be at work in ESCC. MMP9 exhibits substrate specificity for IL-8, and it has been revealed that active MMP9 can make full-length IL-8 up to 10 times more potent by aminoterminal processing in research on neutrophils [52]. In addition to MMP9′s function itself, induced and strengthened IL-8 may also contribute to ESCC progression (Figure 7). Reis et al., reported the co-expression of IL-8 and MMP9 in bladder carcinoma, although the impact of their interactions on cancer progression was not discussed [47]. This point requires further investigation.

S100A8/A9, which we previously reported to be an important factor that facilitates ESCC cell migration and invasion, exhibited only a partial inducible effect on MMP9 secretion from ESCC cells (Figure 7). Considering these incomplete roles of IL-8 and S100A8/A9 in MMP9 production, unknown mechanisms appear to have the key to MMP9 upregulation.

We acknowledge several limitations in the present study. First, in vitro experiments were limited to 2D culture systems. However, actual cancer tissues form a 3D structure, and therefore, experimental models should more closely resemble such real architectures than 2D systems. Furthermore, cellular responses to certain experimental treatments, such as gene expression knockdown or treatments with exogenous agents, may vary in 2D and 3D culture environments. For example, *GDF15* knockdown strongly suppresses *HAMP* mRNA (encoding hepcidin protein) in 3D culture but does not affect *HAMP* transcription in the 2D culture of an MCF-7 (breast carcinoma) cell line [53]. Hence, co-culture systems or IL-8 treatment may yield different results in a 3D culture system or an animal model. Second, the number of cases used in IHC was small. A larger sample size would have further clarified the clinical importance of MMP9 (and IL-8) in ESCC. Finally, this study mainly focused on advanced ESCC. However, with the recent advances and spread of endoscopic therapies, ESCC has been treated in its earlier stages. In future studies, we plan to prepare an early ESCC case series and investigate cancer cell-TAM interactions, which could be useful for the treatment or pathological diagnosis.

## 5. Conclusions

In the present study, we demonstrated that the interaction between ESCC cells and macrophages promotes MMP9 production in ESCC cells using a direct co-culture system. Our findings show that cancer cell MMP9 expression is significantly correlated with M2-like macrophage infiltration and poor clinical outcomes in human ESCC tissues on IHC analysis. Notably, MMP9 expression in cancer stromal cells did not relate to poor clinical features. To our best knowledge, this is the first report to suggest that MMP9 expression in cancer nests and stroma has different impacts on clinical outcomes in ESCC. Our results suggest that MMP9 may be a potential target for molecular therapy and a useful prognostic marker for ESCC.

## Figures and Tables

**Figure 1 cancers-15-02987-f001:**
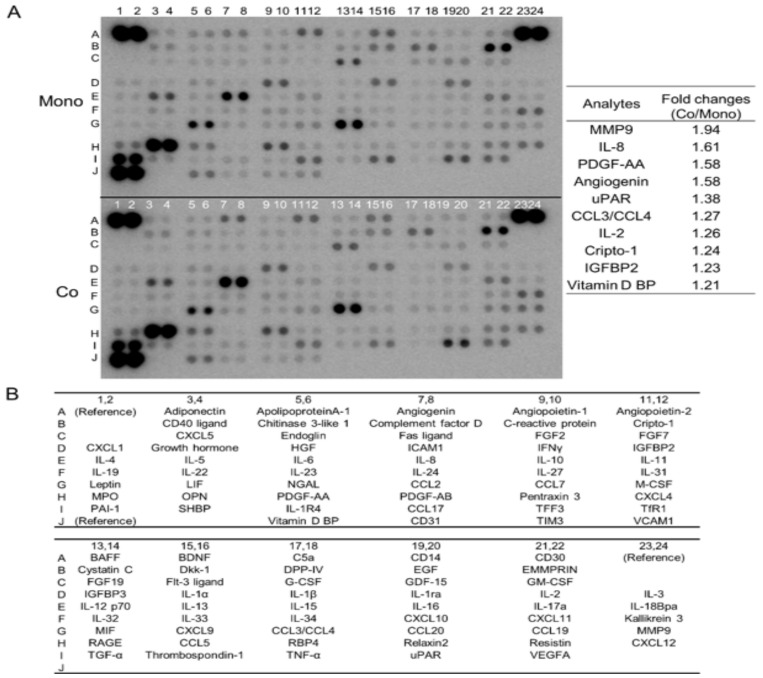
Cytokine array analysis shows increased MMP9 secretion from TE-11 cells after direct co-culture with macrophages. (**A**): Cytokine array analysis of culture supernatants from mono- (upper array) and co-cultured (lower array) TE-11 cells. The densities of spots were analyzed using the ImageJ 1.53t and the top 10 increased fold changes in densities are listed on the right. (**B**): Coordinates of the cytokine array.

**Figure 2 cancers-15-02987-f002:**
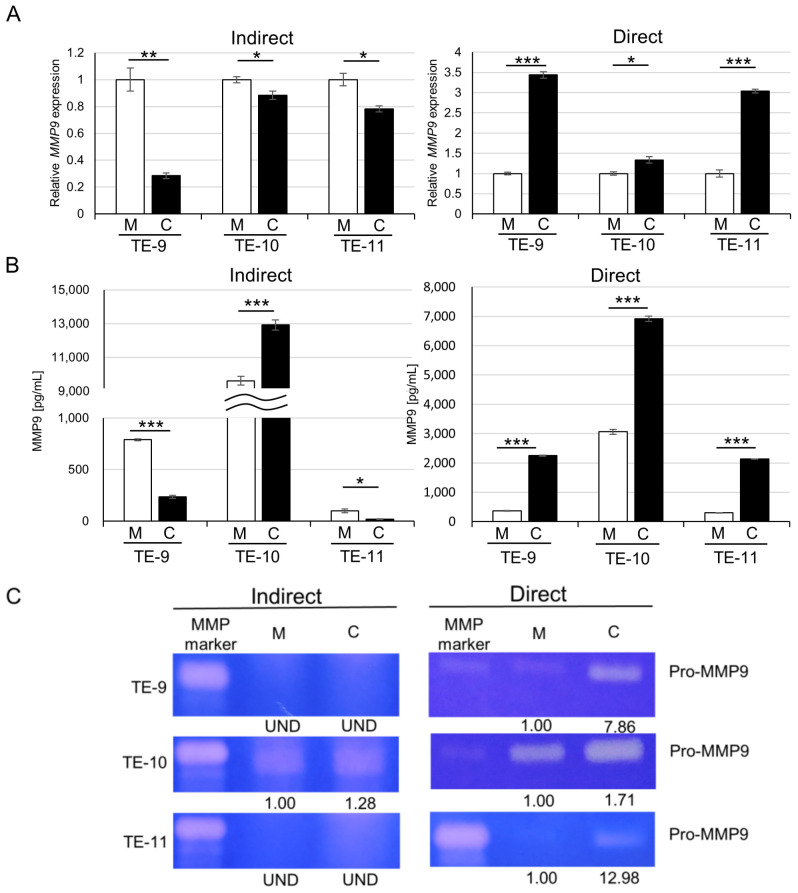
Direct co-culture with macrophages induces MMP9 gene expression and protein secretion in ESCC cells, whereas indirect co-culture does not. (**A**): qPCR revealed upregulation of MMP9 gene transcription in ESCC cells by direct co-culture with macrophages but not by indirect co-culture. (**B**): ELISA quantified MMP9 in culture supernatants of mono- and indirectly or directly co-cultured ESCC cell lines. MMP9 secretion from ESCC cells was elevated by the direct co-culture but not by the indirect co-culture, except for TE-10. (**C**): Gelatin zymography showed increased MMP9 secretion by ESCC cells directly co-cultured with macrophages, and the gelatinolytic activity of MMP9 was preserved in the culture supernatants. Band densities were calculated using the ImageJ 1.53t and expressed as relative values. The band density of monocultured conditions in each cell line was set as 1.00. The indirect co-culture system was performed in 6-well plates, while the direct co-culture system was conducted in 10-cm dishes. Data are exhibited as mean ± SEM; * *p* < 0.05, ** *p* < 0.01, *** *p* < 0.001; M, monocultured; C, co-cultured; UND, undetected.

**Figure 3 cancers-15-02987-f003:**
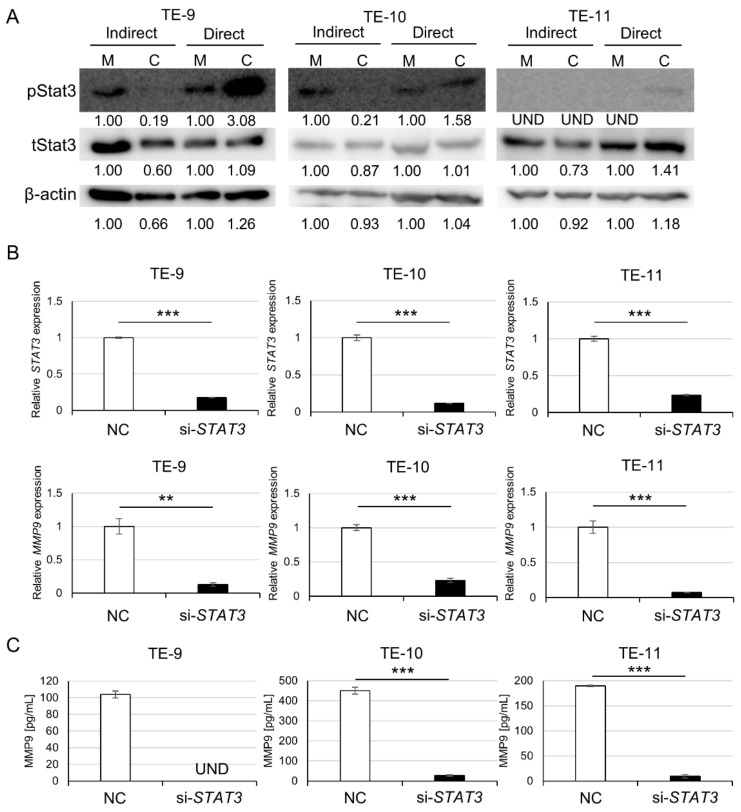
Direct co-culture with macrophages triggers Stat3 phosphorylation, and Stat3 signal pathway may contribute to MMP9 upregulation in ESCC cells. (**A**): Western blot for phosphorylated (p) Stat3 demonstrated increased Stat3 phosphorylation by the direct co-culture but not by the indirect co-culture. (**B**): qPCR showed that STAT3 and MMP9 gene transcriptions in ESCC cells were significantly suppressed by STAT3 knockdown. (**C**): ELISA determined that MMP9 protein secretion from ESCC cells was significantly downregulated by STAT3 knockdown. Band densities were calculated using the ImageJ 1.53t and expressed as relative values. The band density of monocultured conditions in each cell line was set as 1.00. pStat3 was not detected in TE-11 cells of monocultured counterparts and after the indirect co-culture. pStat3, phosphorylated-Stat3; tStat3, total-Stat3; M, monocultured; C, co-cultured; NC, siRNA of negative control; si-STAT3, siRNA against STAT3; UND, undetected. Data are presented as mean ± SEM; UND: undetected; ** *p* < 0.01, *** *p* < 0.001.

**Figure 4 cancers-15-02987-f004:**
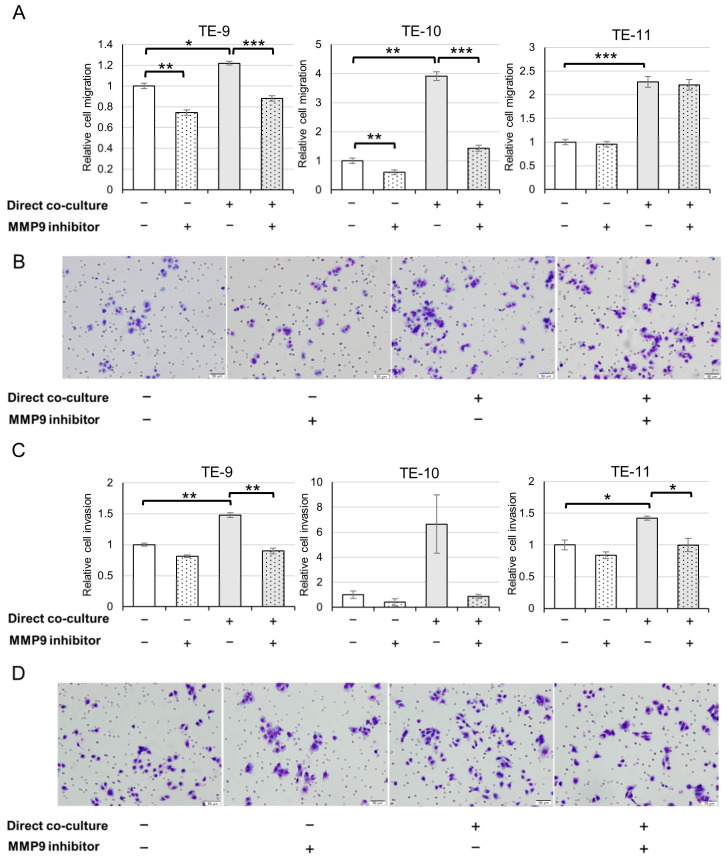
Direct co-culture with macrophages enhances migration and invasion of ESCC cells, which is abrogated by an MMP9 Inhibitor. (**A**): Transwell migration assays revealed that the direct co-culture increased ESCC cell migration. Moreover, the increase was nullified by the MMP9 inhibitor ab142180, except in the assay using TE-11. (**B**): Representative images in the transwell migration assay of TE-11. (**C**): Transwell invasion assays showed that the direct co-culture increased ESCC cell invasion. As in migration assays, increased invasion was significantly abrogated by ab142180 in the assays of TE-9 and TE-11 but not in TE-10, although an inhibitory trend was observed. (**D**): Representative images in the transwell invasion assay of TE-11. Data are exhibited as mean ± SEM; * *p* < 0.05, ** *p* < 0.01, *** *p* < 0.001.

**Figure 5 cancers-15-02987-f005:**
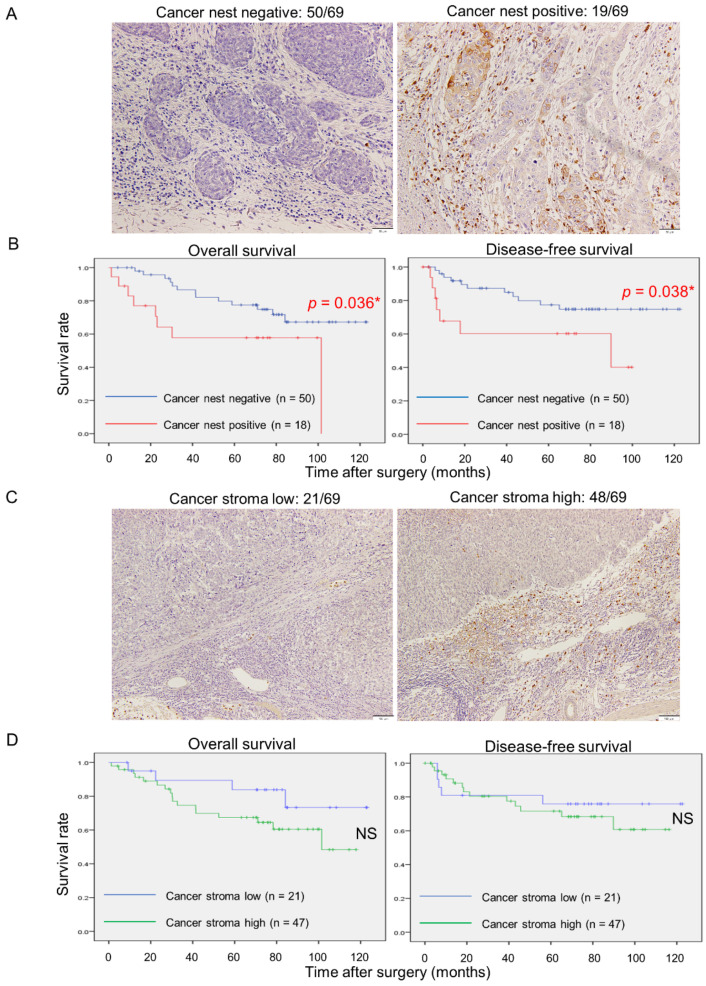
MMP9 immunoreactivity of cancer cells at the invasive front, but not stromal cells, in human ESCC tissues significantly correlates with poorer clinical outcomes. (**A**): Representative images of MMP9 IHC at the invasive front of ESCC tissues. Positive staining was defined as stronger staining than surrounding fibroblasts or lymphocytes. Left, negative (50 out of total 69 cases); Right, positive (19 out of total 69 cases). Scale bars: 50 μm. (**B**): Kaplan–Meier survival analyses of overall (Left) and disease-free survival (Right) based on MMP9 expression of cancer cells at the invasive front. (**C**): Representative images of MMP9 IHC in the cancer stroma of ESCC tissues. Stromal MMP9 expression was classified as low (Left, 21 out of total 69 cases) or high (Right, 48 out of total 69 cases) expressions based on the intensity of adjacent non-neoplastic tissue. Scale bars: 100 μm. (**D**): Kaplan–Meier survival analyses of overall (Left) and disease-free survival (Right) based on MMP9 expression of cancer stromal cells. Data were compared with the log-rank test; * *p* < 0.05; NS, not significant.

**Figure 6 cancers-15-02987-f006:**
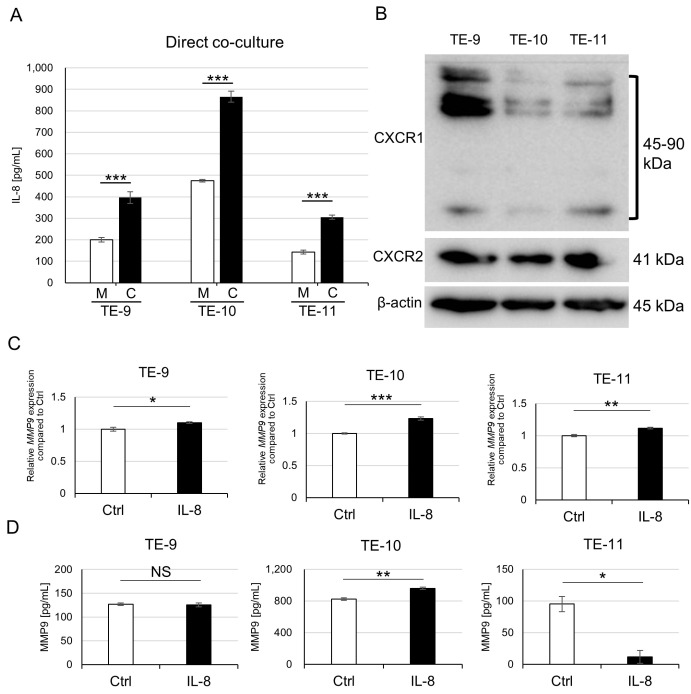
IL-8 as a partial inducer of MMP9 in ESCC cells. (**A**): ELISA quantification of IL-8 in culture supernatants of ESCC cells after direct co-culture with macrophages. Directly co-cultured ESCC cells secreted more IL-8 than monocultured counterparts. (**B**): Western blot revealed the expression of known IL-8 receptors, CXCR1 (45 kDa, monomer; 80–90 kDa, dimer and their glycosylated forms) and CXCR2 (41 kDa) in TE-9, TE-10, and TE-11 cell lines. (**C**,**D**): Treatment with rhIL-8 (100 ng/mL, for 24 h) upregulated *MMP9* mRNA expression in all three ESCC cell lines (**C**) but only triggered MMP9 secretion from TE-10 (**D**). Data are expressed as mean ± SEM; * *p* < 0.05, ** *p* < 0.01, *** *p* < 0.001. NS, not significant.

**Figure 7 cancers-15-02987-f007:**
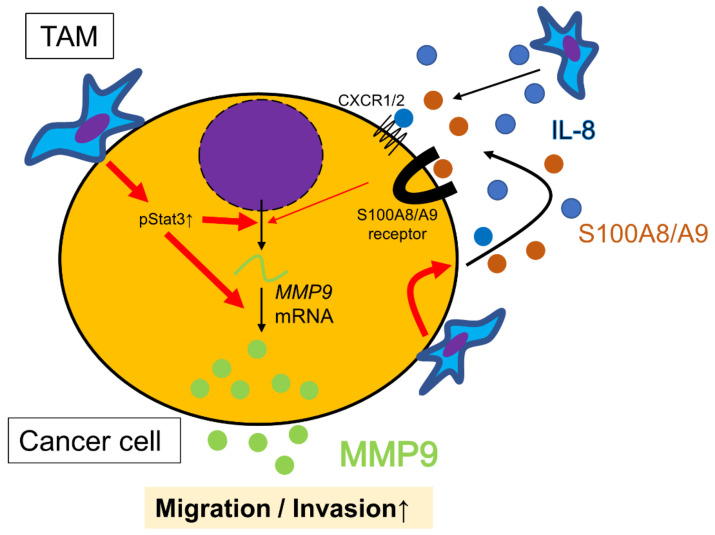
Schematic model of the interactions between ESCC cells and tumor-associated macrophages (TAMs) from the viewpoint of MMP9 regulation in ESCC cells. Direct contact between ESCC cells and TAMs strongly upregulates MMP9 transcription and secretion, promoting migration and invasion of ESCC cells. TAMs also release IL-8, which indirectly induces *MMP9* gene transcription. In addition, the direct contact between ESCC cells and TAMs induces the secretion of IL-8 and S100A8/A9 from ESCC cells, contributing to their migration, invasion and MMP9 production in autocrine or paracrine manner. Collectively, direct contact more effectively promotes ESCC cell motility and invasion than indirect interaction alone.

**Table 1 cancers-15-02987-t001:** Correlations between clinicopathological parameters and MMP9 expression in cancer nests at the invasive front or stromal cells.

		Number of Cases	Cancer Nest Expression of MMP9 ^a^		*p*-Value	Stromal Expression of MMP9 ^b^		*p*-Value
			Negative (n = 50)	Positive (n = 19)		Low (n = 21)	High (n = 48)	
Age								
	<65	32	24	8	0.661	10	22	0.891
	≥65	37	26	11		11	26	
Histological grade ^c^								
CIS + WDSCC		15	12	3	0.460	5	10	0.783
MDSCC + PDSCC		54	38	16		16	38	
Depth of invasion ^c^								
	≤T1	49	39	10	0.038 *	15	34	0.960
	≥T2	20	11	9		6	14	
Lymphatic vessel invasion ^c^								
	Negative	37	29	8	0.237	13	24	0.362
	Positive	32	21	11		8	24	
Blood vessel invasion ^c^								
	Negative	43	33	10	0.306	15	28	0.302
	Positive	26	17	9		6	20	
Lymph node metastasis ^c^								
	Negative	41	32	9	0.209	14	27	0.417
	Positive	28	18	10		7	21	
Stage ^d^								
	≤I	38	30	8	0.182	13	25	0.450
	≥II	31	20	11		8	23	
CD68 positive cells ^e^								
	Low	35	30	5	0.012 *	11	24	0.856
	High	34	20	14		10	24	
CD163 positive cells ^e^								
	Low	34	30	4	0.004 **	11	23	0.733
	High	35	20	15		10	25	
CD204 positive cells ^e^								
	Low	34	33	1	<0.001 ***	13	21	0.165
	High	35	17	18		8	27	

^a^ A cancer nest expression of MMP9 was evaluated at the invasive front and determined whether negative or positive. ^b^ A stromal expression of MMP9 was evaluated in ESCC stroma of invasive lesion and divided into low- and high-groups. ^c^ According to the Japanese Classification of Esophageal Cancer [16,17]. CIS, carcinoma in situ/high-grade intraepithelial neoplasia; WDSCC, well-differentiated squamous cell carcinoma; MDSCC, moderately differentiated squamous cell carcinoma; PDSCC, poorly differentiated squamous cell carcinoma; T1, tumor invades mucosa or submucosa; T2, tumor invades muscularis propria. ^d^ According to the TNM classification by UICC [18]. ^e^ The cases were divided into low- and high-groups according to the median values of CD68, CD163, or CD204 positive macrophage numbers within the ESCC invasive lesions [6]. Data were analyzed with a Chi-square test. * *p* < 0.05, ** *p* < 0.01, *** *p* < 0.001.

**Table 2 cancers-15-02987-t002:** Univariate and multivariate analyses for disease-free survival of clinicopathological parameters, markers of infiltrating tumor-associated macrophages, and MMP9 expression of cancer cells at the invasive front.

		Univariate Analysis		Multivariate Analysis		
		Number of Cases	*p*-Value	HR	95% CI	*p*-Value
Age						
	<65	32	0.377			
	≥65	36				
Histological grade ^a^						
	CIS + WDSCC	15	0.481			
	MDSCC + PDSCC	53				
Depth of tumor invasion ^a^						
	≤T1	48	<0.001 ***	4.547	0.752–27.502	0.010 *
	≥T2	20				
Lymphatic vessel invasion ^a^						
	Negative	37	0.001 **	1.659	0.332–8.298	0.711
	Positive	31				
Blood vessel invasion ^a^						
	Negative	43	0.030 *	0.865	0.260–2.883	0.983
	Positive	25				
Lymph node metastasis ^a^						
	Negative	43	<0.001 ***	0.886	0.089–8.813	0.061
	Positive	25				
Stage ^b^						
	≤I	38	<0.001 ***			
	≥II	30				
CD68 positive cells ^c^						
	Low	33	0.009 **			
	High	35				
CD163 positive cells ^c^						
	Low	34	0.049 *			
	High	34				
CD204 positive cells ^c^						
	Low	34	0.002 **	0.252	0.037–1.703	0.502
	High	34				
Cancer nest MMP9 ^d^						
	Negative	50	0.038 *	4.845	1.399–16.681	0.026 *
	Positive	18				

^a^ According to the Japanese Classification of Esophageal Cancer [16,17]. CIS, carcinoma in situ/high-grade intraepithelial neoplasia; WDSCC, well-differentiated squamous cell carcinoma; MDSCC, moderately differentiated squamous cell carcinoma; PDSCC, poorly differentiated squamous cell carcinoma; T1, tumor invades mucosa or submucosa; T2, tumor invades muscularis propria. ^b^ According to the TNM classification by UICC [18]. ^c^ The cases were divided into low- and high-groups according to the median values of CD68, CD163, or CD204 positive macrophage numbers within the ESCC invasive lesions [6]. ^d^ A cancer nest expression of MMP9 was evaluated at the invasive front and determined whether negative or positive. Univariate analyses multivariate analyses were performed by Kaplan–Meier method with log-rank test and by Cox regression analyses, respectively. HR, hazard ratio; CI, confidence interval; * *p* < 0.05, ** *p* < 0.01, *** *p* < 0.001.

## Data Availability

The data presented in this study are available on request from the corresponding author.

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
