# Peer review of "Matrix Metalloproteinase 9 Induced in Esophageal Squamous Cell Carcinoma Cells via Close Contact with Tumor-Associated Macrophages Contributes to Cancer Progression and Poor Prognosis"

_cancers, 2023, doi:10.3390/cancers15112987_

Round 1
Reviewer 1 Report
In this manuscript, Tsukamoto et. al. used a co-culture paradigm to study the effect of Tumor-Associated Macrophages on cancer cells. They have suggested that direct interaction of esophageal squamous cell carcinoma (ESCC) cells with PBMC-derived macrophages are necessary for the activation of MMP9 expression in ESCC cells. The rationale for investigation on MMP9 is not well established in the introduction. Using the same experimental paradigm, the authors have previously reported S100A8/9 upregulation in ESCC cells (reference 11), which were suggested to mediate many cancer promoting phenotypes in these ESCC cells. The finding on the requirement of the direct cell to cell interaction for the MMP9 gene activation is a novel aspect of the manuscript. However, how the direct cell (macrophage)-to-cell (ESCC) interaction actives the signaling cascade in ESCC is not addressed. Moreover, the reported S100A8/9 mediated effect on cancer cells were not tested on MMP9 expression. Overall, the manuscript needs significant in-depth investigation.
Major concerns:
1. Authors have previously shown that S100A6/A9 mediate most of the similar biological phenotypes of macrophages-tumor cells coculture (reference 11). The results shown in this manuscript could be due to the reported S100A8/A9 effect. This was not tested. Authors should silence or knockout S100A8/A9 from the cancer cells in order to separate its effect from MMP9, and vice versa.
2. Table S2: It is not clear why authors did not find MMP9 as an upregulated gene (Log2 ratio (Co/Mono)= 4.12) from the same dataset ((GSE174796) in their earlier publication (reference #11) where they have reported S100A9/S100A8 upregulation whose Log2 ratio (Co/Mono) were 2.53 and 2.30, respectively.
3. Line 300, and Figure 3A: Authors mentioned that they have performed “western blot of multiple signaling molecules” however only STAT3 data is shown in Fig 3A. What were the other pathways tested, and what was the rational to test STAT3?
4. Figure 3B: Since authors have reported AKT/p38-MAPK pathway activation in the same culture condition (Ref 11), involvement of this pathway should be tested in the activation of STAT3, MMP9 and IL-8.
5. Since macrophages expresses high MMP9, authors should test whether they have some contaminating macrophages after EpCAM selection in the direct co-cultured ESCC cells which might lead to high MMP9 in the direct co-culture sets.
6. Figure 2A-B: Panel-A is redundant with Fig 2B and also misleading since the quantitative difference in MMP9 are not seen in TE-9 and TE-10 cells most probably due to saturation of PCR. For example, in panel A, MMP9 downregulation is clearly seen in TE-11 indirect coculture compared to monoculture, however, panel B suggest the difference is more prominent in TE-9 cells. Since panel B represents more quantitative data, I suggest removing panel A.
7. Figure 3B: this panel is also redundant with Fig 3C.
8. Figure 5A: MMP9 expression in the “cancer nest positive” panel image is mostly on the stromal cells- presumably macrophages or neutrophils which expresses high amount of MMP9- while “cancer nest negative” panel image does not show any staining in stromal cells which may indicate lack of immune cell (macrophage/neutrophils) infiltration into the tissue. The prognostic value (Fig 5B,D) could be explained by immune cell infiltration alone rather than MMP9 expression on the tumor cells.
9. Figure 6C: The changes in MMP9 mRNA level are modest. Are they consistence across repeated experiments?
10. Figure 6D: TE-10 seems to be increasing MMP9 expression upon IL8 treatment. How do authors reconcile this data with decreased MMP9 secretion in IL-8 treated TE-11?
Minor:
11. Figure 6B: Label lanes.
12. Figure 2B-C: Why MMP9 expression is decreased in indirect co-culture?
Reviewer 2 Report
The manuscript is very interesting and very well presented. The authors elegantly describe the involvement of the interaction between ESCC cells and macrophages promotes MMP9 production in ESCC cells using a direct co-culture system. They also demonstrated that MMP9 expression in cancer cells is significantly correlated with M2-like macrophage infiltration and poor clinical outcomes in human ESCC tissues. Another interesting data presented is that MMP9 expression in cancer stromal cells is not related to poor clinical characteristics. I believe these data are the first to demonstrate that MMP9 expression in cancer nests and stromal has different impacts on clinical outcomes in ESCC.
Anyway, interesting data and very well presented. The Material and Method section is well written and the discussion section is favored with the results presented by the group.
Reviewer 3 Report
Tsukamoto et al. attempt to explore the cell-cell interactions between TAMs and ESCC to better understand the tumour microenvironment that could be linked with poor prognosis. In recent, association of TAMs with cancer cells are emerging phenomenon and explored appreciably. However, the role of MMP9 and STAT3 in TAMs and ESCC is presented by the authors.
The following major points may help to achieve better impact of this paper.
1. The authors states that “The crucial role of tumor-associated macrophages (TAMs) in disease progression 19 of various types of tumors has been well established” and hence, authors are encouarged to make strong rationale to explore and demonstrate role of MMP9 and TAMs in ESCC.
2. The authors should discuss the prospects of real tumour tissue environment where many cellular types co-exist, However, in co-culture of TAMs and ESCCs only these two cell types will establish cell-cell communications.
3. The authors are encouraged to explore the cell-cell interactions in 3D models of TAMs and ESCC and this will make stronger claims on the proposed mechanisms.
4. A discussion on the TAMs, ESCC and epigenetic processes such as lactylation could be extended. Since, many target proteins such as histone, STAT3 are explored as site of lactylation in the context of TAMs and ESCCs.
5. Future preclinical and clinical implications need to be strengthened.
Moderate changes are required.
Round 2
Reviewer 1 Report
Revised manuscript addressed most of the comments and has been sufficiently improved.